# Minimax Regret for Cascading Bandits

**Daniel Vial**
UT Austin & UIUC
dvial@utexas.edu

**Sujay Sanghavi**
UT Austin & Amazon
sanghavi@mail.utexas.edu

**Sanjay Shakkottai**
UT Austin
sanjay.shakkottai@utexas.edu

**R. Srikant**
UIUC
rsrikant@illinois.edu

## Abstract

Cascading bandits is a natural and popular model that frames the task of learning to rank from Bernoulli click feedback in a bandit setting. For the case of unstructured rewards, we prove matching upper and lower bounds for the problem-independent (i.e., gap-free) regret, both of which strictly improve the best known. A key observation is that the hard instances of this problem are those with small mean rewards, i.e., the small click-through rates that are most relevant in practice. Based on this, and the fact that small mean implies small variance for Bernoullis, our key technical result shows that variance-aware confidence sets derived from the Bernstein and Chernoff bounds lead to optimal algorithms (up to log terms), whereas Hoeffding-based algorithms suffer order-wise suboptimal regret. This sharply contrasts with the standard (non-cascading) bandit setting, where the variance-aware algorithms only improve constants. In light of this and as an additional contribution, we propose a variance-aware algorithm for the structured case of linear rewards and show its regret strictly improves the state-of-the-art.

## 1 Introduction

The cascading click model describes users interacting with ranked lists, such as search results or online advertisements [Craswell et al., 2008]. In this model, there is a set of items $[L] = \{1, \ldots, L\}$. The user is given a list of $K$ items $\mathbf{A} = (\mathbf{a}_1, \ldots, \mathbf{a}_K)$, sequentially examines the list, and clicks on the first attractive item (if any). If a click occurs, the user leaves without examining the subsequent items. It is assumed that the $e$-th item has an attraction probability $\bar{w}(e)$ (which we also call the mean reward), and that the random (Bernoulli) clicks are conditionally independent given $\mathbf{A}$.

Cascading bandits, introduced concurrently by Kveton et al. [2015a] and Combes et al. [2015], are a sequential learning version of the model where the mean rewards $\{\bar{w}(e)\}_{e=1}^{L}$ are initially unknown. At each round $t \in [n]$, the learner chooses an action, which is a list of items $\mathbf{A}_t = (\mathbf{a}_1^t, \ldots, \mathbf{a}_K^t)$. As in the basic click model, the user scans the list and clicks on the first attractive item (if any). Thus, if the $\mathbf{C}_t$-th item is clicked, the learner knows that the user was not attracted to $\{\mathbf{a}_k^t\}_{k=1}^{\mathbf{C}_t-1}$ but was attracted to $\mathbf{a}_{\mathbf{C}_t}^t$. However, the learner receives *no* feedback on the items $\{\mathbf{a}_k^t\}_{k=\mathbf{C}_t+1}^{K}$ that the user did not examine before leaving. The objective for the learner is to choose the sequence $\{\mathbf{A}_t\}_{t=1}^{n}$ to maximize the expected number of clicks, or equivalently, minimize the regret defined in (1).

This work provides problem-independent (i.e., gap-free) regret bounds for cascading bandits that strictly improve the state-of-the-art. In the case of unstructured rewards, our results provide the first minimax-optimal regret bounds (up to log terms). Our key insight is that, compared to the standard bandit problem, the reward variance plays an outsized role in the gap-free analysis. In particular, we show that *for cascading bandits, the worst-case problem instances are those with low mean rewards –*

Table 1: Problem-independent upper bounds and minimax lower bounds for cascading bandits with $L$ total items, $K$ recommended items, horizon $n$, and feature dimension $d$. The tabular columns assume unstructured mean reward. The linear column assumes reward is linear in unit-norm features and does not suppress any $L$ dependencies in the $\tilde{O}(\cdot)$ notation. See Appendix A for other related work.

| Paper | Tabular case | | Linear case |
| | Upper bound | Lower bound | Upper bound |
| --- | --- | --- | --- |
| Zong et al. [2016] | none | none | $\tilde{O}(\sqrt{nd^2K^2})$ |
| Wang and Chen [2017] | $\tilde{O}(\sqrt{nLK})$ | none* | none |
| Lattimore et al. [2018] | $\tilde{O}(\sqrt{nLK^3})$ | none* | none |
| Li and Zhang [2018] | none | none | $\tilde{O}(\sqrt{nd^2K})^{\S}$ |
| Zhong et al. [2021] | $\tilde{O}(\sqrt{nLK})$ | $\Omega(\sqrt{nL/K})^{\dagger}$ | $\tilde{O}(\sqrt{nd^2K^3\min\{d,\log L\}})$ |
| Kveton et al. [2022] | $\tilde{O}(\sqrt{nLK})^{\ddagger}$ | none | none |
| Ours | $\tilde{O}(\sqrt{nL})$ (Thm 2) | $\Omega(\sqrt{nL})^{\dagger}$ (Thm 1) | $\tilde{O}(\sqrt{nd(d+K)})^{\S}$ (Thm 4) |

*These papers contain minimax lower bounds but for click models that are distinct from cascading bandits.
†These bounds assume $L$ is large compared to $K$ and $n$ is large compared to $L$ and $K$.
‡This bound holds for a Bayesian notion of regret and includes a dependence on the prior not shown above.
§These bounds assume $n$ is large compared to $d$ and $K$ to suppress some additive $o(\sqrt{n})$ terms.

namely, $\bar{w}(e) \le 1/K$, where the variance $\bar{w}(e)(1-\bar{w}(e))$ is also small. We emphasize that these worst-case instances are not pathological; rather, they model the low click-through rates that prevail in practice [Richardson et al., 2007]. Further, we argue that *adapting to low variance is crucial to cope with the worst-case instances*. Put differently, algorithms should be *variance-aware*, i.e., more exploitative when the variance is small. We provide the intuition behind this key insight in Section 1.2 and show how to formalize it with a proof sketch in Section 5.

More specifically, our first goal is to establish upper bounds on the problem-independent regret (i.e., the maximum over $\bar{w}$) for cascading bandit algorithms, as well as minimax lower bounds (i.e., the infimum over algorithms of their problem-independent regret). As shown in Table 1, the tightest existing such bounds are $\tilde{O}(\sqrt{nLK})$ and $\Omega(\sqrt{nL/K})$, respectively. What is surprising is that, not only is there a gap between the bounds, but they *increase* and *decrease* in $K$, respectively. In other words, the following fundamental question is unresolved: *as $K$ grows, does the problem become harder* (as suggested by the upper bound) *or easier* (as suggested by the lower bound)?

As discussed by Zhong et al. [2021], this question lacks an obvious answer. On the one hand, larger $K$ means that the learner needs to identify more good items, which hints at a harder problem. On the other hand, the learner receives more feedback as $K$ grows, which intuitively makes the problem easier. As we show later, variance-aware algorithms are the key to resolving this tradeoff.

In addition to this basic version of the model – hereafter, the *tabular case*, where no structure is assumed for the mean rewards – we are also interested in the *linear case*, where $\bar{w}(e) = \langle \phi(e), \theta \rangle$ for some known feature map $\phi : [L] \to \mathbb{R}^d$ and unknown parameter vector $\theta \in \mathbb{R}^d$. A second goal of this work is to apply our insights from the tabular case to the linear one, in hopes of improving the best known upper bound $\tilde{O}(\sqrt{nd^2K})$ (see Table 1).

## 1.1 Main contributions

**Tabular case.** First, we show no algorithm can achieve $o(\sqrt{nL})$ regret uniformly across $\bar{w}$, assuming $L$ is large compared to $K$ and $n$ is large compared to $L$ (see Theorem 1). Next, we consider three algorithms: `CascadeKL-UCB`, `CascadeUCB-V`, and `CascadeUCB1` (the first and third are due to Kveton et al. [2015a]; `CascadeUCB-V` is new). All three rank the $L$ items using upper confidence bounds (UCBs) and choose $\mathbf{A}_t$ as the $K$ highest ranked items. They differ in the choice of UCB. As the names suggest, `CascadeKL-UCB` and `CascadeUCB-V` use KL-UCB [Garivier and Cappé, 2011, Maillard et al., 2011, Cappé et al., 2013] and UCB-V [Audibert et al., 2009], respectively. Both are variance-aware, in the sense that their respective UCBs are derived from the Chernoff and Bernstein inequalities. We show that both algorithms have near-optimal regret $\tilde{O}(\sqrt{nL})$ for any $\bar{w}$ (see Theorem 2). In contrast, `CascadeUCB1` relies on the Hoeffing-style UCB1 [Auer et al., 2002], and we show that it suffers suboptimal regret $\Omega(\sqrt{nLK})$ on some $\bar{w}$ (see Theorem 3).

In summary, we prove (i) the minimax regret is $\tilde{\Theta}(\sqrt{nL})$ for cascading bandits, (ii) the variance-aware algorithms `CascadeKL-UCB` and `CascadeUCB-V` are minimax-optimal up to log terms, and (iii) the variance-unaware algorithm `CascadeUCB1` is decidedly *sub*optimal. Moreover, note from Table 1 that we strictly improve *both* the upper and lower bounds for this problem.

**Discussion.** There are two surprising aspects to these results. First, the minimax bound $\tilde{\Theta}(\sqrt{nL})$ shows that (in a worst-case sense) the number of recommended items $K$ plays *no role*.[1] In other words, the aforementioned tradeoff (identifying more good items but receiving more feedback) is perfectly balanced when the correct algorithm (e.g., `CascadeKL-UCB`) is employed.

The second (and arguably more surprising) aspect is that `CacadeKL-UCB` and `CascadeUCB-V` are optimal but `CascadeUCB1` is not. This stands in contrast to the standard $L$-armed bandit problem, where the analogous algorithms `KL-UCB`, `UCB-V`, and `UCB1` all achieve the minimax regret $\tilde{\Theta}(\sqrt{nL})$, and the main advantage of the former two is only to improve the constants in the gap-dependent bounds. We discuss the intuition behind this contrast in Section 1.2.

**Linear case.** Motivated by these findings, we also propose a variance-aware algorithm for the linear case called `CascadeWOFUL`.[2] `CascadeWOFUL` proceeds in two steps. First, we use the Hoeffding-style UCBs of Abbasi-Yadkori et al. [2011] to upper bound the mean rewards $\bar{w}(e)$, and thus the Bernoulli variances $\bar{w}(e)(1 - \bar{w}(e))$, with high probability. Second, we use these Hoeffding UCBs as proxies for the true variances in the `WOFUL` algorithm of Zhou et al. [2021], which computes a variance-weighted estimate of $\theta$ that enjoys Bernstein-style concentration. In Theorem 4, we show the regret of `CascadeWOFUL` is $\tilde{O}(\sqrt{nd(d + K)})$ for large $n$, which improves existing bounds by a factor of at least $\sqrt{\min\{d, K\}}$ (see Table 1).

## 1.2 Why do variance-aware algorithms succeed but variance-unaware algorithms fail?

To answer this question, we focus on the tabular case and contrast the standard $L$-armed bandit setting with the cascading one. We first recall how the $\tilde{O}(\sqrt{nL})$ bound for $L$-armed bandits is derived. For simplicity, we restrict to instances with $\bar{w}(1) = p$ and $\bar{w}(2) = \cdots = \bar{w}(L) = p - \Delta$ for some $p \in (0, 1)$ and $\Delta \in (0, p)$. In this case, `UCB1` plays each of the $L - 1$ suboptimal items $1/\Delta^2$ times, up to constants and log factors [Auer et al., 2002]. Each such play costs regret $\Delta$, for a total regret that scales as $L/\Delta$. Alternatively, regret can simply be bounded by $\Delta n$, since the $n$ plays incur at most $\Delta$ regret each. Combining the bounds gives $\min\{L/\Delta, \Delta n\}$. The worst case occurs when the two bounds are equal, i.e., when $\Delta = \sqrt{L/n}$, which implies $\sqrt{nL}$ regret.

In contrast, `UCB-V` plays each suboptimal item $\sigma^2/\Delta^2$ times (in an order sense), where $\sigma^2 \leq p$ is the variance of the Bernoulli$(p - \Delta)$ reward [Audibert et al., 2009]. Therefore, the argument of the previous paragraph shows that regret grows as $\min\{Lp/\Delta, \Delta n\}$. Here the worst case occurs when $p$ is non-vanishing and $\Delta = \sqrt{L/n}$, which gives the same regret scaling as `UCB1`. A similar argument holds for `KL-UCB`, because the number of plays grows as $1/d(p - \Delta, p)$ [Cappé et al., 2013] and one can show $d(p - \Delta, p) = \Omega(\Delta^2/p)$ (see Claim 1 in Appendix G).

The analysis is more complicated for cascading bandits, because regret is nonlinear in the mean rewards and the amount of feedback is random. To oversimplify things, we draw an analogy with the above and assume $\bar{w}(1) = \cdots = \bar{w}(K) = p$ and $\bar{w}(K + 1) = \cdots = \bar{w}(L) = p - \Delta$. In this case, `CascadeUCB1` similarly plays the $L - K$ suboptimal items $1/\Delta^2$ times each, which costs $L/\Delta$ regret when $L \gg K$. However, we can no longer bound regret by $\Delta n$, because the total number of plays depends on the random number of items the user examines at each round, which is roughly $\min\{1/p, K\}$ (the mean of Geometric$(p)$ random variable, truncated to the maximum $K$). Thus, the $\Delta n$ bound inflates to $\Delta n \min\{1/p, K\}$, which gives $\min\{L/\Delta, \Delta n \min\{1/p, K\}\}$ regret. For $p \leq 1/K$ and $\Delta = \sqrt{L/(nK)}$, this yields the best known bound $\sqrt{nLK}$. We emphasize that, unlike the previous paragraph, *the worst case here occurs when $p$ (i.e., the click-through rate) is small*.

On the other hand, the analogous bound for `CascadeUCB-V` and `CascadeKL-UCB` scales as $\min\{Lp/\Delta, \Delta n \min\{1/p, K\}\}$. Crucially, the factor of $p$ in the first term – which arises due to the variance-aware nature of the algorithms – offsets the factor $1/p$ in the second term. Thus, in the hard

---

[1] The $\tilde{O}(\cdot)$ notation hides $\log K$ terms, but they can be bounded by $\log L$ while retaining $\tilde{O}(\sqrt{nL})$ regret.
[2] `WOFUL` stands for weighted optimism in the face of uncertainty for linear bandits.

case $p \leq 1/K$, the bound becomes $\min\{L/(\Delta K), \Delta n K\}$. Here the worst case is $\Delta = \sqrt{L/(nK^2)}$, which yields the minimax regret $\sqrt{nL}$ that we establish in Theorems 1 and 2.

## 2  Preliminaries

In this section, we precisely formulate our problem. A cascading bandit instance is defined by the triple $(L, K, \bar{w})$, where $L \in \mathbb{N}$ is the total number of items, $K \in [L] = \{1, \ldots, L\}$ is the number of items the learner displays to the user at each round, and $\bar{w} \in [0, 1]^L$ is the vector of attraction probabilities. We define a sequential game as follows. At each round $t \in [n]$, the learner chooses an *action* $\mathbf{A}_t = (\mathbf{a}_1^t, \ldots, \mathbf{a}_K^t)$, where $\mathbf{a}_k^t \in [L]$ and $\mathbf{a}_k^t \neq \mathbf{a}_{k'}^t$ when $k \neq k'$ (i.e., $\mathbf{A}_t$ is an ordered list of $K$ distinct items). The user sequentially examines this list, clicks on the first item that attracts them, and stops examining items after clicking. Mathematically, we denote the first attractive item by $\mathbf{C}_t = \inf\{k \in [K] : \mathbf{w}_t(\mathbf{a}_k^t) = 1\}$, where $\mathbf{w}_t(e) \sim \text{Bernoulli}(\bar{w}(e))$ for each $e \in [L]$. If no item is clicked, i.e., if $\mathbf{w}_t(\mathbf{a}_k^t) = 0$ for all $k$, we set $\mathbf{C}_t = \infty$. Note the learner only observes the realizations corresponding to the items that the user examined, i.e., $\{\mathbf{w}_t(\mathbf{a}_k^t) : k \in [\min\{\mathbf{C}_t, K\}]\}$.

We denote by $\mathcal{H}_t = \cup_{s=1}^{t-1}\{\mathbf{A}_s\} \cup \{\mathbf{w}_s(\mathbf{a}_k^s) : k \in [\min\{\mathbf{C}_s, K\}]\}$ the *history* of actions and observations before time $t$.[3] We let $\mathbb{P}_t(\cdot) = \mathbb{P}(\cdot|\mathcal{H}_t \cup \{\mathbf{A}_t\})$ and $\mathbb{E}_t[\cdot] = \mathbb{E}[\cdot|\mathcal{H}_t \cup \{\mathbf{A}_t\}]$ denote conditional probability and expectation given the history and current action. In the cascading bandit model, it is assumed that the feedback $\{\mathbf{w}_t(\mathbf{a}_k^t)\}_{k=1}^K$ (i.e., the presence or absence of clicks) is conditionally independent given $\mathcal{H}_t$ and $\mathbf{A}_t$. Therefore, the conditional click probability is

$$\mathbb{P}_t(\mathbf{C}_t < \infty) = 1 - \mathbb{P}_t(\mathbf{C}_t = \infty) = 1 - \mathbb{P}_t(\cap_{k=1}^K\{\mathbf{w}_t(\mathbf{a}_k^t) = 0\}) = 1 - \prod_{k=1}^K(1 - \bar{w}(\mathbf{a}_k^t)).$$

Mappings from $\mathcal{H}_t$ to $\mathbf{A}_t$ are called *policies*. Let $\Pi$ be the set of all policies. Given an instance $\bar{w} \in [0, 1]^L$, a policy $\pi \in \Pi$, and a horizon $n \in \mathbb{N}$, the expected number of clicks is

$$\mathbb{E}_{\pi,\bar{w}}\left[\sum_{t=1}^n \mathbb{1}(\mathbf{C}_t < \infty)\right] = \mathbb{E}_{\pi,\bar{w}}\left[\sum_{t=1}^n \mathbb{P}_t(\mathbf{C}_t < \infty)\right] = \mathbb{E}_{\pi,\bar{w}}\left[\sum_{t=1}^n \left(1 - \prod_{k=1}^K(1 - \bar{w}(\mathbf{a}_k^t))\right)\right],$$

where $\mathbb{1}$ is the indicator function. Let $A^* = (a_1^*, \ldots, a_K^*)$ be any action $A = (a_1, \ldots, a_K)$ that maximizes the click probability $1 - \prod_{k=1}^K(1 - \bar{w}(a_k))$. Our goal is to minimize *regret*, which is the expected difference in the number of clicks between $\pi$ and the policy that always plays $A^*$, i.e.,

$$R_{\pi,\bar{w}}(n) = \mathbb{E}_{\pi,\bar{w}}\left[\sum_{t=1}^n \left(\prod_{k=1}^K(1 - \bar{w}(\mathbf{a}_k^t)) - \prod_{k=1}^K(1 - \bar{w}(a_k^*))\right)\right]. \tag{1}$$

## 3  Results for the tabular case

We can now state our tabular results (the proofs are discussed in Section 5). First, we have a minimax lower bound showing no algorithm can achieve $o(\sqrt{nL})$ uniformly over the mean rewards $\bar{w}$.

**Theorem 1.** *Suppose $N \triangleq L/K \in \{4, 5, \ldots\}$ and $n \geq L$. Then for any policy $\pi \in \Pi$, there exists a mean reward vector $\bar{w} \in [0, 1]^L$ such that $R_{\pi,\bar{w}}(n) = \Omega(\sqrt{nL})$.*

**Remark 1.** *The proof of Theorem 1 is essentially a reduction to Lattimore et al. [2018]'s lower bound for the so-called document-based click model. Their proof and ours both use the assumption $L/K \in \mathbb{N}$ to simplify the analysis, which involves partitioning the $L$ items into $K$ subsets of size $L/K$ each. When $L/K \notin \mathbb{N}$, one of the subsets will have fewer items, which makes the analysis more cumbersome; however, this does not fundamentally alter either result.*

**Remark 2.** *The theorem also requires $L \geq 4K$, which is not very restrictive since $L \gg K$ in typical applications. However, this assumption does eliminate an interesting analytical regime, namely, when $K \to L$. We conjecture the minimax lower bound is $\Omega(\sqrt{n(L - K)})$ in this case, since any algorithm obtains zero regret when $K = L$ and there are no suboptimal items.*

---

[3]Our notation mostly follows Kveton et al. [2015a], but we clarify that their definition of $\mathcal{H}_t$ includes $\mathbf{A}_t$.

We next consider the algorithms `CascadeKL-UCB` and `CascadeUCB1` from Kveton et al. [2015a], along with a new one called `CascadeUCB-V`. All follow a similar template, which is given in Algorithm 1. This is a natural generalization of upper confidence bound (UCB) algorithms from the standard $L$-armed bandit setting. At each round $t \in [n]$, it computes UCBs $\mathbf{U}_t(e)$ in a manner to be specified shortly, then chooses $\mathbf{A}_t$ as the $K$ items with the highest UCBs (in order of UCB). After observing the click feedback $\mathbf{C}_t$, the algorithm increments the number of observations $\mathbf{T}_t(e)$ and updates the empirical mean $\hat{\mathbf{w}}_{\mathbf{T}_t(e)}(e)$ for each item $e$ that the user examined.

---

**Algorithm 1:** General UCB algorithm for tabular cascading bandits

---

Initialize number of observations $\mathbf{T}_t(e) = 0$ and empirical mean $\hat{\mathbf{w}}_0(e) = 0$ for each $e \in [L]$
**for** $t = 1, \ldots, n$ **do**
    Compute $\mathbf{U}_t(e)$ for each $e \in [L]$ (by (2), (3), and (4) for `CascadeKL-UCB`, `CascadeUCB-V`,
    and `CascadeUCB1`, respectively)
    **for** $k = 1, \ldots, K$ **do**
        Let $\mathbf{a}_k^t = \arg\max_{e \in [L] \setminus \{\mathbf{a}_i^t\}_{i=1}^{k-1}} \mathbf{U}_t(e)$ be the item with the $k$-th highest UCB
    Play $\mathbf{A}_t = (\mathbf{a}_1^t, \ldots, \mathbf{a}_K^t)$ and observe $\mathbf{C}_t = \inf\{k \in [K] : \mathbf{w}_t(\mathbf{a}_k^t) = 1\}$ (where $\inf \emptyset = \infty$)
    Let $\mathbf{T}_t(e) = \mathbf{T}_{t-1}(e)$ for each $e \in [L]$
    **for** $k = 1, \ldots, \min\{\mathbf{C}_t, K\}$ **do**
        $e \leftarrow \mathbf{a}_k^t, \mathbf{T}_t(e) \leftarrow \mathbf{T}_{t-1}(e) + 1, \hat{\mathbf{w}}_{\mathbf{T}_t(e)}(e) \leftarrow (\mathbf{T}_{t-1}(e)\hat{\mathbf{w}}_{\mathbf{T}_{t-1}(e)}(e) + \mathbb{1}(\mathbf{C}_t = k))/\mathbf{T}_t(e)$

---

For `CascadeKL-UCB`, the UCBs $\mathbf{U}_t(e)$ are computed as follows:

$$\mathbf{U}_t(e) = \max\{u \in [0, 1] : d(\hat{\mathbf{w}}_{\mathbf{T}_{t-1}(e)}(e), u) \leq \log(f(t))/\mathbf{T}_{t-1}(e)\}, \qquad (2)$$

where $f(t) = t(\log t)^3$ and $d(p, q) = p \log(p/q) + (1-p) \log((1-p)/(1-q))$ is the relative entropy between Bernoullis with means $p, q \in [0, 1]$. The set in (2) is a confidence interval for $\bar{w}(e)$, which is derived from the Chernoff bound. For `CascadeUCB-V`, the UCBs are instead given by

$$\mathbf{U}_t(e) = \hat{\mathbf{w}}_{\mathbf{T}_{t-1}(e)}(e) + \sqrt{4\hat{\mathbf{v}}_{\mathbf{T}_{t-1}(e)}(e) \log(t)/\mathbf{T}_{t-1}(e)} + 6 \log(t)/\mathbf{T}_{t-1}(e), \qquad (3)$$

where $\hat{\mathbf{v}}_s(e) = \hat{\mathbf{w}}_s(e)(1 - \hat{\mathbf{w}}_s(e))$ is the empirical variance from $s$ observations of item $e$. This UCB is derived from the coarser – but crucially, still variance-aware – Bernstein inequality. Finally, for `CascadeUCB1`, the UCBs are derived from the Hoeffding bound and computed as follows:

$$\mathbf{U}_t(e) = \hat{\mathbf{w}}_{\mathbf{T}_{t-1}(e)}(e) + c_{t, \mathbf{T}_{t-1}(e)}, \quad \text{where} \quad c_{t,s} = \sqrt{1.5 \log(t)/s}. \qquad (4)$$

We can now show the variance-aware UCBs are nearly optimal, while `CascadeUCB1` is suboptimal.

**Theorem 2.** *Suppose $\pi$ is `CascadeKL-UCB` or `CascadeUCB-V`, i.e., the policy from Algorithm 1 with the UCBs given by (2) or (3). Then $R_{\pi, \bar{w}}(n) = \tilde{O}(\sqrt{nL})$ for any $\bar{w} \in [0, 1]^L$.*

**Remark 3.** *The reader may wonder why we proposed `CascadeUCB-V`, since the `CascadeKL-UCB` bound is enough to establish the minimax regret. The main reason is to demonstrate that variance-awareness alone (no additional information encoded by `KL-UCB`) is enough to achieve the optimal regret, which helps motivate our linear algorithm. Furthermore, we show empirically in Section 6 that, while `CascadeUCB-V` is inferior to `CascadeKL-UCB` in terms of regret, its closed form nature leads to quicker computation, while still improving the regret of `CascadeUCB1`.*

**Theorem 3.** *Suppose $n \geq \max\{LK, 49K^4\}$, $L \geq 800K$, and $\pi$ is `CascadeUCB1`, i.e., the policy from Algorithm 1 with the UCBs given by (4). Then $R_{\pi, \bar{w}}(n) = \Omega(\sqrt{nLK})$ for some $\bar{w} \in [0, 1]^L$.*

**Remark 4.** *The assumed bounds on $n$ and $L$ simplify the calculations and can be improved (see Remark 9 in Appendix J). The main point of Theorem 3 is to show that, unlike the variance-aware algorithms, `CascadeUCB1` cannot satisfy the conclusion of Theorem 2 for all choices of $n$ and $L$.*

## 4 Results for the linear case

In light of the previous section, we seek a variance-aware algorithm for the linear case. Our method is based on the `WOFUL` algorithm of Zhou et al. [2021], which was designed for the standard linear

bandit setting (the case $K = 1$). In this section, we review `WOFUL`, discuss how to overcome its limitations in the cascading setting, explain our `CascadeWOFUL` algorithm, and bound its regret.

**Existing algorithm.** In Section 4.3 of their work, Zhou et al. [2021] consider the following problem. As above, there is a set of items $[L]$, a known feature map $\phi : [L] \to \mathbb{R}^d$, and an unknown parameter vector $\theta \in \mathbb{R}^d$. Successive plays of $e \in [L]$ give i.i.d. rewards with mean $\bar{w}(e) = \langle \phi(e), \theta \rangle$ and variance upper bounded by $\sigma_e^2$. Thus, at each round $t \in [n]$, the learner chooses $\mathbf{a}^t \in [L]$ and receives a random reward $r_t = \langle \phi(\mathbf{a}^t), \theta \rangle + \eta_t$, where $\mathbb{E}[\eta_t | \mathbf{a}^t] = 0$ and $\mathbb{E}[\eta_t^2 | \mathbf{a}^t] \le \sigma_{\mathbf{a}^t}^2$. For this setting, the authors proposed the `WOFUL` algorithm, which is based on the (unweighted) `OFUL` algorithm [Abbasi-Yadkori et al., 2011]. At each round $t \in [n]$, `WOFUL` chooses the item

$$\mathbf{a}^t = \arg\max_{e \in [L]} \left( \langle \phi(e), \hat{\theta}_t \rangle + \alpha \|\phi(e)\|_{\mathbf{\Lambda}_t^{-1}} \right), \tag{5}$$

where $\alpha > 0$ is an exploration parameter, $\|x\|_B = \sqrt{x^\mathsf{T} B x}$ is the norm induced by a positive definite matrix $B$, and $\hat{\theta}_t$ is the regularized and variance-weighted least-squares estimate given by

$$\hat{\theta}_t = \mathbf{\Lambda}_t^{-1} \sum_{s=1}^{t-1} \phi(\mathbf{a}^s) r_s / \sigma_{\mathbf{a}^s}^2, \quad \text{where} \quad \mathbf{\Lambda}_t = I + \sum_{s=1}^{t-1} \phi(\mathbf{a}^s) \phi(\mathbf{a}^s)^\mathsf{T} / \sigma_{\mathbf{a}^s}^2. \tag{6}$$

To gain some intuition, we assume momentarily that $d = L$ and $\phi(e)$ is the $e$-th standard basis vector, i.e., the vector with 1 in the $e$-th coordinate and 0 elsewhere. In this case, one can easily calculate

$$\langle \phi(e), \hat{\theta}_t \rangle = \frac{\sum_{s \in [t-1] : \mathbf{a}^s = e} r_s / \sigma_e^2}{1 + \mathbf{T}_{t-1}(e) / \sigma_e^2}, \quad \|\phi(e)\|_{\mathbf{\Lambda}_t^{-1}} = \sqrt{\frac{1}{1 + \mathbf{T}_{t-1}(e) / \sigma_e^2}}.$$

Therefore, for large $\mathbf{T}_{t-1}(e)$ (large enough that $1 + \mathbf{T}_{t-1}(e) / \sigma_e^2 \approx \mathbf{T}_{t-1}(e) / \sigma_e^2$), we have

$$\langle \phi(e), \hat{\theta}_t \rangle + \alpha \|\phi(e)\|_{\mathbf{\Lambda}_t^{-1}} \approx \frac{\sum_{s \in [t-1] : \mathbf{a}^s = e} r_s}{\mathbf{T}_{t-1}(e)} + \alpha \sqrt{\frac{\sigma_e^2}{\mathbf{T}_{t-1}(e)}}.$$

Similar to `UCB-V` (3), the right side of this equation is the empirical mean plus an exploration bonus that grows with the variance upper bound $\sigma_e^2$ and decays in $\mathbf{T}_{t-1}(e)$. Hence, the term inside the $\arg\max$ in (5) can be interpreted as a Bernstein-style UCB. The analysis of this UCB relies on a novel concentration inequality for vector-valued martingales [Zhou et al., 2021, Theorem 2], which is a Bernstein analogue of the Hoeffding-style bound due to Abbasi-Yadkori et al. [2011].

**Limitations.** The fact that `WOFUL` (roughly) generalizes `UCB-V` is promising. However, computing (6) requires knowledge of the variance upper bounds $\sigma_e^2$, and nontrivial bounds on the variance are rarely available in practice. We sidestep this issue with three simple observations: (i) cascading bandits only involve Bernoulli rewards, (ii) for Bernoulli rewards, variances are upper bounded by means, and (iii) these means can be learned efficiently since they are linearly-parameterized. This suggests the following algorithm: first, compute Hoeffding-style UCBs; second, treat these UCBs as upper bounds for the true means, and thus the true variances, in `WOFUL`.

**Proposed algorithm.** Algorithm 2 formalizes this approach. It contains three steps. First, step 1 defines Hoeffding-style UCBs $\mathbf{U}_{t,H}$ as in Abbasi-Yadkori et al. [2011]. Next, step 2 uses $\mathbf{U}_{t,H}$ as an upper bound for the variance and computes the Bernstein-style UCBs $\mathbf{U}_{t,B}$ as in `WOFUL`. Finally, step 3 chooses $\mathbf{A}_t$ as the $K$ items with the highest $\mathbf{U}_{t,B}$, analogous to Algorithm 1.

Two technical clarifications are in order. First, observe that in step 1, we clip the variance bound $\mathbf{U}_{t,H}$ below by $1/K$. We do so to ensure that $\mathbf{\Lambda}_{t,B}$ (which inverts $\mathbf{U}_{t,H}$) remains bounded. Additionally, we note the choice $1/K$ is precisely motivated by Section 1.2, which shows this is a critical threshold for the small click-through rate. Second, $\mathbf{\Lambda}_{t,B}$ uses the regularizer $KI$. This is to ensure that the regularizer is large enough compared to the summands $\phi(\mathbf{a}_k^s) \phi(\mathbf{a}_k^s)^\mathsf{T} / \mathbf{U}_{s,H}(\mathbf{a}_k^s)$, which scale as $K$ in the worst case where the variance upper bound $\mathbf{U}_{s,H}(\mathbf{a}_k^s)$ is clipped to $1/K$.

**Remark 5.** *Zong et al. [2016] proposed a `WOFUL`-style cascading algorithm called `CascadeLinUCB` (see Appendix C), but they set the variance upper bound to fixed $\sigma^2 > 0$. In fact, they remark "ideally, $\sigma^2$ should be the variance of the observation noises," which `CascadeWOFUL` essentially learns.*

**Remark 6.** *Appendix B contains an improved version of `CascadeWOFUL`. It is more efficient (for example, the inverses are iteratively updated via Sherman-Morrison), satisfies the same theoretical guarantee as Algorithm 2, and includes some tweaks that improve performance in practice. The tradeoff is that Algorithm 2 is simpler to explain, which is why we prefer it for the main text.*

**Algorithm 2:** `CascadeWOFUL` for linear cascading bandits

---

**Input:** exploration parameters $\{\alpha_{t,H}, \alpha_{t,B}\}_{t=1}^n$

**for** $t = 1, \ldots, n$ **do**

 **Step 1:** Define the (clipped) Hoeffding-style UCBs

$$\mathbf{U}_{t,H}(\cdot) = \max\left\{\langle\phi(\cdot), \hat{\theta}_{t,H}\rangle + \alpha_{t,H}\|\phi(\cdot)\|_{\mathbf{\Lambda}_{t,H}^{-1}}, 1/K\right\}, \quad \text{where}$$

$$\hat{\theta}_{t,H} = \mathbf{\Lambda}_{t,H}^{-1}\sum_{s=1}^{t-1}\sum_{k=1}^{\min\{\mathbf{C}_s,K\}}\phi(\mathbf{a}_k^s)\mathbf{w}_s(\mathbf{a}_k^s), \quad \mathbf{\Lambda}_{t,H} = I + \sum_{s=1}^{t-1}\sum_{k=1}^{\min\{\mathbf{C}_s,K\}}\phi(\mathbf{a}_k^s)\phi(\mathbf{a}_k^s)^\mathsf{T}$$

 **Step 2:** Define the Bernstein-style UCBs

$$\mathbf{U}_{t,B}(\cdot) = \langle\phi(\cdot), \hat{\theta}_{t,B}\rangle + \alpha_{t,B}\|\phi(\cdot)\|_{\mathbf{\Lambda}_{t,B}^{-1}}, \quad \text{where}$$

$$\hat{\theta}_{t,B} = \mathbf{\Lambda}_{t,B}^{-1}\sum_{s=1}^{t-1}\sum_{k=1}^{\min\{\mathbf{C}_s,K\}}\frac{\phi(\mathbf{a}_k^s)\mathbf{w}_s(\mathbf{a}_k^s)}{\mathbf{U}_{s,H}(\mathbf{a}_k^s)}, \quad \mathbf{\Lambda}_{t,B} = KI + \sum_{s=1}^{t-1}\sum_{k=1}^{\min\{\mathbf{C}_s,K\}}\frac{\phi(\mathbf{a}_k^s)\phi(\mathbf{a}_k^s)^\mathsf{T}}{\mathbf{U}_{s,H}(\mathbf{a}_k^s)}$$

 **Step 3:** Let $\mathbf{a}_k^t = \arg\max_{e \in [L]\setminus\{\mathbf{a}_i^t\}_{i=1}^{k-1}}\mathbf{U}_{t,B}(e)$ be the item with the $k$-th highest UCB,
 play $\mathbf{A}_t = (\mathbf{a}_1^t, \ldots, \mathbf{a}_K^t)$, observe $\mathbf{C}_t = \inf\{k \in [K] : \mathbf{w}_t(\mathbf{a}_k^t) = 1\}$ (as in Algorithm 1)

---

**Regret bound.** We can now state our main result for the linear case. Here $\mathscr{B}_d(M) = \{x \in \mathbb{R}^d : \|x\|_2 \leq M\}$ denotes the Euclidean ball of radius $M > 0$ in $\mathbb{R}^d$.[4]

**Theorem 4.** *Suppose $\bar{w} \in [0,1]^L$, $\phi : [L] \to \mathscr{B}_d(1)$, and $\theta \in \mathscr{B}_d(1)$ satisfy $\bar{w}(e) = \langle\phi(e), \theta\rangle$ for all $e \in [L]$. Let $\pi$ be the policy of Algorithm 2 with inputs $\alpha_{t,H} = \sqrt{d\log(1 + tK/d) + 2\log(n)} + 1$ and $\alpha_{t,B} = 8\sqrt{d\log(1 + tK/d)\log(n^3K)} + 4\sqrt{K}\log(n^3K) + \sqrt{K}$. Then*

$$R_{\pi,\bar{w}}(n) = \tilde{O}\left(\min\left\{\sqrt{nd(d+K)} + n^{\frac{1}{6}}d^{\frac{7}{6}}(d+K)^{\frac{1}{2}}K^{\frac{1}{3}}, \sqrt{n}\max\{d,K\}\min\{d,K\}^{1/3}\right\}\right).$$

Note this bound is completely independent of the number of items $L$. For large $n$ (typically the case of interest), it becomes $\tilde{O}(\sqrt{nd(d+K)})$, which improves the best known bound of $\tilde{O}(\sqrt{nd^2K})$ [Li and Zhang, 2018]. The theorem also establishes $\tilde{O}(\sqrt{n}\max\{d,K\}\min\{d,K\}^{1/3})$ regret *uniformly* in $n$ (i.e., without additive $o(\sqrt{n})$ terms which may dominate for small $n$), which improves the best known uniform-$n$ bound of $\tilde{O}(\sqrt{n}dK)$ [Zong et al., 2016]. See Table 1 for more details.

## 5  Overview of the analysis

We next discuss the key ideas behind our proofs. The details are deferred to Appendices E-J.

**Theorem 2 proof.** For simplicity, we assume the arms are ordered by their means, i.e., $\bar{w}(1) \geq \cdots \geq \bar{w}(L)$. Under this assumption, the optimal action is $[K]$. We call those items *optimal* and $[L]\setminus[K]$ *suboptimal*. We let $\mathcal{E}_t$ be the "bad event" that the empirical and true means differ substantially at time $t$, $\bar{\mathcal{E}}_t$ its complement, $G_{e,e^*,t}$ the event that suboptimal $e > K$ was chosen in favor of optimal $e^* \leq K$ and subsequently examined by the user, and $\Delta_{e,e^*} = \bar{w}(e^*) - \bar{w}(e)$ the *reward gap*. Then as in Appendix A.1 of Kveton et al. [2015a], we "linearize" regret as follows:

$$R_{\pi,\bar{w}}(n) \leq \mathbb{E}\left[\sum_{t\leq n}\sum_{e>K}\sum_{e^*\leq K}\Delta_{e,e^*}\mathbb{1}(\bar{\mathcal{E}}_t, G_{e,e^*,t})\right] + \sum_{t\leq n}\mathbb{P}(\mathcal{E}_t). \tag{7}$$

The second term is small due to concentration. For the first term, define $\Delta = \bar{w}(K)\sqrt{L/n}$ and assume $n \gg L$ (so $\Delta \ll \bar{w}(K)$).[5] For each $e > K$, let $\mathcal{K}_\Delta(e) = \{e^* \leq K : \Delta_{e,e^*} \leq \Delta\}$ be the

---

[4]Our results hold with minor modification if $\phi : [L] \to \mathscr{B}_d(M_1)$ and $\theta \in \mathscr{B}_d(M_2)$ for general $M_1, M_2 > 0$. However, as in prior work, the modified algorithm needs to know an upper bound on $M_2$.

[5]The full proof addresses the cases $n \ggg L$ and $\bar{w}(K) = 0$ (here we implicitly assume the latter to invert $\Delta$).

optimal items with small gap relative to $e$ and $\bar{\mathcal{K}}_\Delta(e) = [K] \setminus \mathcal{K}_\Delta(e)$ the other optimal items. Then

$$\sum_{e^* \leq K} \Delta_{e,e^*} \mathbb{1}(\bar{\mathcal{E}}_t, G_{e,e^*,t}) \leq \Delta \sum_{e^* \in \mathcal{K}_\Delta(e)} \mathbb{1}(G_{e,e^*,t}) + \sum_{e^* \in \bar{\mathcal{K}}_\Delta(e)} \Delta_{e,e^*} \mathbb{1}(\bar{\mathcal{E}}_t, G_{e,e^*,t}).$$

Therefore, we can upper bound the first term in (7) by

$$\Delta \sum_{t \leq n} \mathbb{E}\left[\sum_{e>K} \sum_{e^* \in \mathcal{K}_\Delta(e)} \mathbb{1}(G_{e,e^*,t})\right] + \mathbb{E}\left[\sum_{t \leq n} \sum_{e>K} \sum_{e^* \in \bar{\mathcal{K}}_\Delta(e)} \Delta_{e,e^*} \mathbb{1}(\bar{\mathcal{E}}_t, G_{e,e^*,t})\right]. \tag{8}$$

For the first term in (8), the inner double summation is the number of $e > K$ that were chosen in favor of $e^* \in \mathcal{K}_\Delta(e)$, and subsequently examined by the user, at round $t$. Denote this number by $\mathbf{O}_t$. Recall $\bar{w}(e^*) \geq \bar{w}(K)$ (by the assumed ordering) and $\Delta \ll \bar{w}(K)$, so for $e^* \in \mathcal{K}_\Delta(e)$, we have

$$\bar{w}(e) = \bar{w}(e^*) - \Delta_{e,e^*} \geq \bar{w}(e^*) - \Delta \geq \bar{w}(K) - \Delta \gtrsim \bar{w}(K).$$

Hence, $\mathbf{O}_t$ is bounded by the number of items $e$ with $\bar{w}(e) \approx \bar{w}(K)$ that the user examined. Since the user stops examining at the first attractive item, this number is dominated by a Geometric$(\bar{w}(K))$ random variable. Therefore, the first term in (8) is $\tilde{O}(\Delta n / \bar{w}(K)) = \tilde{O}(\sqrt{nL})$.

The second term in (8) accounts for choosing $e$ instead of $e^*$ when $\Delta_{e,e^*} \geq \Delta$ and the empirical means are concentrated. Building upon the intuition of Section 1.2, we exploit the variance-awareness of KL-UCB and UCB-V (and use $\bar{w}(e) \leq \bar{w}(K)$ by the assumed ordering) to bound this term by $\tilde{O}(L\bar{w}(e)(1 - \bar{w}(e))/\Delta) \leq \tilde{O}(L\bar{w}(K)/\Delta)$. Thus, by choice of $\Delta$, this term is $\tilde{O}(\sqrt{nL})$ as well.

**Theorem 4 proof.** We decompose regret similar to (7) and (8), though with different choices of $\mathcal{E}_t$ and $\Delta$.[6] To bound $\mathbb{P}(\mathcal{E}_t)$, we use the aforementioned result of Abbasi-Yadkori et al. [2011] to show that the Hoeffding UCBs $\mathbf{U}_{t,H}$ upper bound the variances with high probability, then prove a guarantee for the least-squares estimate $\hat{\theta}_{t,B}$ using the Bernstein-style bound from Zhou et al. [2021]. We bound the first term in (8) using the exact same logic as the tabular case. The second term has a more complicated analysis, but (as in the tabular case) it amounts to bounding the number of times that items $\Delta$-far from optimal are chosen when $\hat{\theta}_{t,B}$ is well-concentrated. For this, we adapt techniques from the standard linear bandit setting (the case $K = 1$) to general $K \in [L]$.

**Theorem 1 proof.** We let $\mathcal{H}'_t = \cup_{s=1}^{t-1} \{\mathbf{A}_s, \mathbf{w}_s(\mathbf{a}_1^s), \ldots, \mathbf{w}_s(\mathbf{a}_K^s)\}$ be the *entire* history before time $t$, which includes unobserved rewards $\mathbf{w}_s(\mathbf{a}_k^s), k > \mathbf{C}_s$. We also let $\Pi'$ be the policies that map $\mathcal{H}'_t$ to $\mathbf{A}_t$. Note $\mathcal{H}_t \subset \mathcal{H}'_t$, so $\Pi \subset \Pi'$. For any $\pi \in \Pi'$ and $\bar{w} \in [0,1]^L$, we define

$$R'_{\pi,\bar{w}}(n) = \mathbb{E}_{\pi,\bar{w}}\left[\sum_{t=1}^n \sum_{k=1}^K (\bar{w}(a_k^*) - \bar{w}(\mathbf{a}_k^t))\right], \tag{9}$$

where $a_k^* = \arg\max_{e \in [L] \setminus \{a_i^*\}_{i=1}^{k-1}} \bar{w}(e)$. Then a lower bound linearization analogous to (7) shows that for any $p \in [0,1]$ and $\bar{w} \in [0,p]^L$, $R_{\pi,\bar{w}}(n) \geq (1-p)^{K-1} R'_{\pi,\bar{w}}(n)$, which implies

$$\inf_{\pi \in \Pi} \sup_{\bar{w} \in [0,1]^L} R_{\pi,\bar{w}}(n) \geq \inf_{\pi \in \Pi'} \sup_{\bar{w} \in [0,p]^L} R_{\pi,\bar{w}}(n) \geq (1-p)^{K-1} \inf_{\pi \in \Pi'} \sup_{\bar{w} \in [0,p]^L} R'_{\pi,\bar{w}}(n). \tag{10}$$

If we choose $p = 1$, the $\inf \sup$ at right is the minimax regret for the document-based model that was analyzed by Lattimore et al. [2018] (see Remark 1), but this makes (10) vacuous. On the other hand, by choosing $p = O(1/K)$ (again, the small click-through rate of Section 1.2), so that the term $(1-p)^{K-1}$ in (10) is $\Omega(1)$, we can modify their analysis to prove Theorem 1.

**Theorem 3 proof.** We linearize the regret similar to the proof of Theorem 1, then define a problem instance reminiscent of Section 1.2 and (roughly) follow the intuition for UCB1 therein.

## 6 Experiments

Before closing, we conduct experiments on both synthetic and real data. Some details regarding experimental setup are deferred to Appendix C. Code is available in the supplementary material.

---

[6]In fact, the proofs of Theorems 2 and 4 both rely on a more general gap-free regret decomposition for cascading bandits (Lemma 1 in Appendix F), which to our knowledge is novel and may be of independent interest.

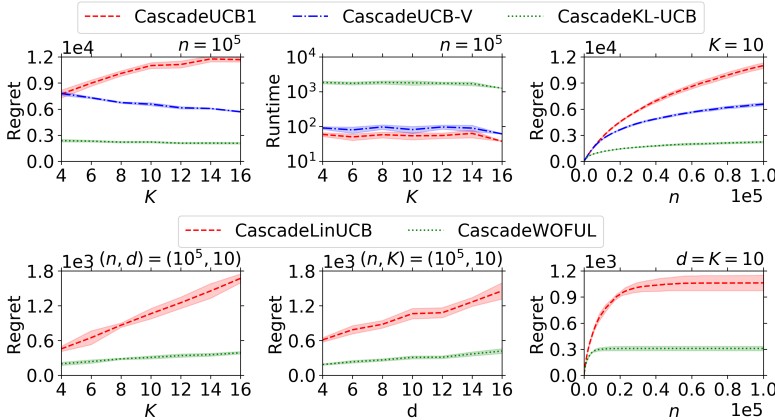

Figure 1: Results for synthetic data (tabular on top, linear on bottom, $L = 100$ in both)

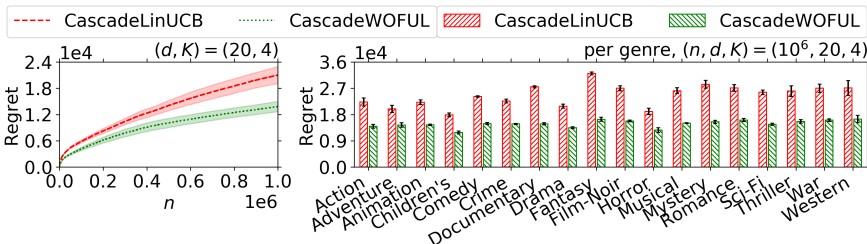

Figure 2: Results for MovieLens data [Harper and Konstan, 2015]

**Synthetic data.** We let $L = 100$ and $K \in \{2i\}_{i=2}^{8}$. For each $K$, we sample $\bar{w}(e)$ uniformly in $[\frac{2}{3K}, \frac{1}{K}]$ for $e \le K$ and in $[0, \frac{1}{3K}]$ for $e > K$. Note this yields a positive gap with the small click-through rate of Section 1.2. The top left plot in Figure 1 shows the regret at $n = 10^5$ for the tabular albums of Section 3 (the shaded regions are the standard deviations across five trials). As predicted by Theorems 2 and 3, the CascadeUCB1 curve grows with $K$, while the variance-aware curves do not. The top middle plot shows the time spent computing UCBs for the same experiment, which confirms the behavior mentioned in Remark 3. For the linear case, we vary $d \in \{2i\}_{i=2}^{8}$, generate the same $\bar{w}$, then compute unit-norm vectors $\theta$ and $\phi(e)$ satisfying $\bar{w}(e) = \langle \phi(e), \theta \rangle$ (see Appendix C). We compare CascadeWOFUL to CascadeLinUCB [Zong et al., 2016], which Li and Zhang [2018] showed has the best existing regret guarantee. As suggested by Theorem 4, the left and middle plots on the bottom of Figure 1 show that our algorithm's regret has superior dependence on $K$ and $d$. The rightmost plots show that regret is sublinear in $n$ for the median values $K = d = 10$.

**Real data.** We replicate the first experiment from Zong et al. [2016] on the MovieLens-1M dataset (`grouplens.org/datasets/movielens/1m/`), which contains user ratings for $L \approx 4000$ movies. In brief, the setup is as follows. First, we use their default choices $d = 20$ and $K = 4$. Next, we divide the ratings into train and test sets based on the user who provided the rating. From the training data and a rank-$d$ SVD approximation, we learn a feature mapping $\phi$ from movies to the probability that a uniformly random training user rated the movie more than three stars. Finally, we run the algorithms as above, except at round $t \in [n]$, we sample a uniformly random user $\mathbf{J}_t$ from the test set and define $\mathbf{w}_t(\mathbf{a}_k^t) = W(\mathbf{J}_t, \mathbf{a}_k^t)$, where $W(j, a) = \mathbb{1}(\text{user } j \text{ rated movie } a \text{ more than 3 stars})$. In other words, instead of the independent Bernoulli clicks of Section 2, we observe the actual feedback of user $\mathbf{J}_t$. We point the reader to Section 4 of Zong et al. [2016] and Appendix C for further details. The left plot of Figure 2 shows that CascadeWOFUL outperforms CascadeLinUCB across $n$, eventually incurring less than 66% of the regret. In addition to this setup from Zong et al. [2016], we reran the experiment while restricting the set of items to movies of a particular genre, for each of 18 genres in the dataset. This is intended to model platforms like Netflix that recommend movies in various categories. The right plot shows that CascadeWOFUL is superior for all genres; for some genres (e.g., fantasy) its regret is about half of CascadeLinUCB's. Moreover, our experiments indicate that CascadeWOFUL

improves `CascadeLinUCB` more dramatically for genres with smaller click-through rates (see Figure 3 and surrounding discussion in Appendix C), which reinforces a key message of this paper.

# 7 Conclusion

In this work, we proved matching upper and lower bounds for the problem-independent regret of tabular cascading bandits and an upper bound for the linear case, all of which improve the best known. Our results suggest some interesting future directions, such as proving minimax lower bounds for the linear case and revisiting Thompson sampling for cascading bandits [Zhong et al., 2021] in light of our variance-aware insight; see Appendix D for details. Finally, we note the paper is theoretical and has no immediate societal impact. Nevertheless, we urge caution for the negative impacts that could arise in practice. For example, our MovieLens experiments involved training on a subset of users, which could cause poor recommendations for demographics underrepresented in the training set.

## Acknowledgments and Disclosure of Funding

This work was partially supported by ONR Grant N00014-19-1-2566, NSF TRIPODS Grant 1934932, NSF Grants CCF 22-07547, CCF 19-34986, CNS 21-06801, 2019844, 2112471, 2107037, the Machine Learning Lab (MLL) at UT Austin, and the Wireless Networking and Communications Group (WNCG) Industrial Affiliates Program. We thank Advait Parulekar for helpful discussions.

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
