# OpenReview forum: "Minimax Regret for Cascading Bandits"
_NeurIPS.cc/2022/Conference — NeurIPS 2022 Accept_

### Official Review · Reviewer_hXD7 · 2022-07-08

**Rating:** 6
**Confidence:** 4
**Soundness:** 3 good
**Presentation:** 3 good
**Contribution:** 3 good

**Summary:**

This work studied the regret minimization in cascading bandits. It closed the gap for the upper and lower bounds on the regret in the tabular setting by developing a variance-aware UCB-based algorithm. It also studied the linear setting and evaluated the algorithms with numerical experiments.

**Questions:**

1. What are the key analytical challenges? These should be described in greater detail in Section 5. The current analytical framework in Section 5 looks quite standard for most cascading bandit algorithms.
2. Is it possible to also provide a lower bound in the linear setting? This may further complete the analysis in cascading bandits.
3. As shown in previous works such as Zhong et al. [2021], Thompson sampling algorithms can also be applied in cascading bandits, why don't the authors compare them in numerical experiments?


**Limitations:**

I don't see any issue regarding societal impact.

**Strengths And Weaknesses:**

Strengths: This work is easy to follow and the claims are generally sound. It improved the existing upper and lower bounds and closed the gap in the tabular setting. It also discussed why involving variance in the design of the algorithm can help. The numerical experiments compared different UCB-type algorithms.

Weaknesses: The key difference between the analysis in this work and previous work should be better elaborated, and more experiments are appreciated. See the "Questions" part for more details.

---

> ### Author Response · Authors · 2022-08-02
> **response to review**
>
> Thanks for the great questions.
>
> Regarding the analysis, we would characterize our contribution as a new insight which allows us to get much tighter results than previously known. In particular, our main insight – which, to our knowledge, is completely new – is that the parameter $\Delta$ in the minimax analysis should be chosen in terms of $p$ (in the notation of Section 1.2) to balance the random number of items $\Delta$-close to optimal that the user examines with the number of plays of items $\Delta$-far from optimal (provided the algorithm is variance-aware). Of course, the discussion in Section 1.2 greatly oversimplifies this to highlight the intuition; more generally, one must choose $\Delta$ in terms of $\bar{w}(K)$ as in Section 5, and figuring out all the details when the mean rewards are less homogeneous (i.e., not all $p$ or $p-\Delta$ as in Section 1.2) requires more work.
>
> Regarding the linear lower bound and Thompson sampling, we didn’t have time to fully address these issues due to the short response period. However, we added Appendix D (and a pointer in the conclusion of the main text) to discuss both issues as important future directions. (Changes are in purple text in the revision.) This includes some tentative ideas on the lower bound and some tentative experiments on Thompson sampling.

---

> > ### Comment · Reviewer_hXD7 · 2022-08-08
> > **Rebuttal is acknowledged**
> >
> > Thanks for the authors for the clarification.

---

### Official Review · Reviewer_HkuS · 2022-07-09

**Rating:** 8
**Confidence:** 4
**Soundness:** 4 excellent
**Presentation:** 4 excellent
**Contribution:** 4 excellent

**Summary:**

The cascading-bandits problem considers a scenario where the agent aims to choose K out of L items, and each item corresponds to an independent Bernoulli random variable. In each iteration, the agent chooses an ordered list of K items, and the underlying variable is revealed in the same order until a 1 is observed. So the overall reward and regret for each action can be written in terms of the product of the probabilities of getting 0 for the K chosen actions.

In prior works, the best known upper bound for the optimal regret given n iterations is increasing in K. The best known lower bound is decreasing in K, leaving an otherwise gap. This work completely resolves this problem for $K\leq L/4$.

**Questions:**

Theorem 1 is stated only for the discrete case of L/K being an integer. As far as I understand, this should imply the same lower bound for any $L/K\geq 4$ (within a constant factor), or even $L/K\geq C$ for general $C>1$. Is there any reason that the integer constraint is imposed in the theorem statement?

**Limitations:**

--

**Strengths And Weaknesses:**

The main strength of this work is the complete resolution of cascading bandits, at least for $K\leq L/4$. Besides the results, the techniques/algorithms proposed in this work are also interesting. Note that when the reward function is linear (i.e., being the sum of expected values for the K items), the minimax regret is known. Cascading bandits have a non-linear regret function, and I imagined the solution of cascading bandits would be something that handles the non-linearity through sophisticated analysis. The presented algorithm uses KL divergence, which is surprisingly clean.

Perhaps one limitation of this work is that the lower bound only holds for $L/K\geq C$ with a constant $C>1$, leaving the minimax regret for the $K\rightarrow L$ regime open. The minimax regret in that regime is known to follow a different rule, and it would be of interest whether the proposed solution or techniques can provide optimal bounds in that regime as well.

---

> ### Author Response · Authors · 2022-08-02
> **response to review**
>
> Thanks for the great questions.
>
> Regarding the integer constraint, it arises from Lattimore et al. 2018’s analysis, on which our proof is based. In both proofs, the constraint is not fundamental; it just makes the analysis cleaner. In more detail, the proofs construct hard instances (as usual for minimax regret analyses) where the $L$ items are partitioned into $K$ subsets of $L/K$ items each, and the $K$ optimal items in each instance are chosen by picking one item from each subset. If $L/K$ is not an integer, this construction will be messier (the last subset will have fewer items), but the proof won’t significantly change (the constant 4 may slightly increase, though).
>
> Regarding the case $K \rightarrow L$, we did not address it because $L$ is typically much larger than $K$ in the relevant applications. We also note that the existing lower bound $\Omega(\sqrt{n L / K})$ from Zhong et al. 2021 makes the same assumption (not stated explicitly, but their proof treats $1-K/L$ as a non-vanishing constant). That all being said, we agree this is an interesting mathematical question. We conjecture that the minimax regret is $\sqrt{n (L-K)}$ in this case, which is the analogue of the multi-armed bandit lower bound $\sqrt{n (L-1)}$, i.e., “root of horizon times number of suboptimal items”. (Note the MAB lower bound is typically written as $\sqrt{n L}$, but more precisely, it is $\sqrt{n (L-1)}$, since any algorithm obtains zero regret when $L=1$ and there are no suboptimal items. Similarly, in cascading bandits, any algorithm obtains zero regret when $L=K$ and there are no suboptimal items.)

---

> > ### Comment · Reviewer_HkuS · 2022-08-08
> > **Reply**
> >
> > The authors' responses are satisfactory, so I would keep the strong acceptance recommendation.
> >
> > Here are some intuitions on why the integer constraint can be removed without cost. First, for any fixed K, the minimax regret should be non-decreasing in L. This is because, for any larger L, a strict subclass of scenarios is that a fraction of items is known to have zero rewards for sure. The minimax regret for this subclass is no greater than the overall minimax regret, and is identical to the overall minimax regret for an alternative set of parameters with the same K and reduced L. So with no additional information required, the presented main theorem itself gives bounds between $\Theta(\sqrt{nK\lceil L/K\rceil})$ and $\Theta(\sqrt{nK\lfloor L/K\rfloor})$ for $L/K\geq 4$, which is $\Theta(\sqrt{nL})$. In this way, the integer issue is removed without having to relax the $1/4$ condition.
> >
> > From the reviewer's perspective, it would be helpful to the reader to add this intuition to the paper.

---

> > > ### Author Response · Authors · 2022-08-08
> > > **thanks for the intuition**
> > >
> > > Thanks for your comment; the intuition is very clear. We will add a remark discussing this.

---

### Official Review · Reviewer_mQJt · 2022-07-11

**Rating:** 7
**Confidence:** 4
**Soundness:** 3 good
**Presentation:** 4 excellent
**Contribution:** 4 excellent

**Summary:**

This paper studies the cascading bandits in both tabular and linear cases. The paper's main contribution is to find out that the hardest problem instances (where means are small) also yield small variance, meaning that the estimation for these instances is naturally more accurate. Based on this observation, the authors use variance-aware algorithms to give improved regret bounds, reducing the $K$-related factors. Specifically, for the tabular case, the authors show that the previous Cascade-KL and Cascade-V algorithms can achieve the improved regret upper bound by a factor of $O(K)$, compared with previous works. The authors also show that the newly proposed bounds are optimal up to log terms, using their newly proposed lower bounds. For the linear case, the authors propose the CascadeWOFUL algorithm and use the mean to replace the unknown variance. The regret bound of CascadeWOFUL improves the existing results by a factor of $O(\sqrt{K\min(d,K)})$. Finally, the experiments show that the proposed algorithms achieve superior performance compared to baselines.

**Questions:**

Please justify the questions proposed in the weakness part.

**Limitations:**

The authors adequately addressed the limitations and potential negative societal impact of their work.

**Strengths And Weaknesses:**

Strength.

1. The current paper makes solid contributions to the cascading bandits and combinatorial multi-armed bandits literature. For the tabular case, both upper and lower bounds strictly improve the existing works. For the linear case, the derived upper bound also significantly improves the existing works.

2. The presentation is very clear, and I enjoy the flow of showing their results. The intuition of the algorithm design and the proof is quite clear and easy to follow. Both the novelty and the challenge of the problem are properly illustrated.

3. Beyond the theoretical analysis, the authors also give promising empirical results, which adds more practical value to their work.

Weakness.
1. I wonder what the gap-dependent regret bound is. I believe the distribution-dependent regret bound is harder to derive, since the gap-independent regret bound can be derived by optimizing the gap $\Delta$ using the distribution-dependent bound. Can the gap-dependent bound be easily shown using the current paper’s analysis?

2. In appendix A, the authors claim Li and Zhang [2018] requires additional assumptions. In fact, the additional assumptions (cluster regularity, user uniformness and item regularity) are to deal with the varying $\theta$. If $\theta$ are the same, they are not needed to give the result of Corollary 3. So I think the authors could double-check their claim.

---

> ### Author Response · Authors · 2022-08-02
> **response to review**
>
> Thanks for the great questions.
>
> Regarding the gap-dependent bounds, our proofs use the known gap-dependent analysis for cascading bandits, along with our novel ideas regarding the role of variance, to obtain tighter gap-free bounds than previously known. We would like to emphasize that extending the gap-dependent ideas to gap-free bounds in this manner is much trickier in cascading bandits than multi-armed bandits (where the $\tilde{O}(\sqrt{n L})$ proof requires just a few more lines – see, e.g., the proof of Theorem 7.2 in Lattimore and Szepesvari 2020). In particular, our main insight – which, to our knowledge, is completely new – is that the parameter $\Delta$ in the minimax analysis should be chosen in terms of $p$ (in the notation of Section 1.2) to balance the random number of items $\Delta$-close to optimal that the user examines with the number of plays of items $\Delta$-far from optimal (provided the algorithm is variance-aware). Of course, the discussion in Section 1.2 greatly oversimplifies this to highlight the intuition; more generally, one must choose $\Delta$ in terms of $\bar{w}(K)$ as in Section 5, and figuring out all the details when the mean rewards are less homogeneous (i.e., not all $p$ or $p-\Delta$ as in Section 1.2) requires more work.
>
> Regarding the Li and Zhang paper, thanks for pointing this out. Their assumptions were stated as part of the model and we didn’t realize they could be relaxed when the model is specialized to our setting. We updated the paper accordingly (changes in purple text).

---

> > ### Comment · Reviewer_mQJt · 2022-08-04
> > **Reply to the authors' response**
> >
> > Thanks for the revision on the comparison with the Li and Zhang paper, the current comparison looks good to me. As for the gap-dependent bound, I still believe the gap-dependent bound is harder (though I am not 100\% sure about cascading bandit setting here). However, considering the context of gap-free regret bound (which is the main focus of this paper), the current work indeed makes significant contributions to the literature, so I keep my score unchanged and vote for acceptance of the current paper.

---

### Official Review · Reviewer_GvNk · 2022-07-11

**Rating:** 7
**Confidence:** 3
**Soundness:** 3 good
**Presentation:** 3 good
**Contribution:** 3 good

**Summary:**

This paper revisits the cascading bandits problem where the decision maker must choose an ordered list of $K$ distinct items (from a universe of $L \geqslant K$ items) at each round $t\in\left\lbrace 1,...,n \right\rbrace$ to display to users arriving sequentially. Each item in the universe has a latent attraction probability; users are homogeneous, scan the ordered list displayed to them in a fixed order (left-to-right), and click on the first item that attracts them (subsequent items are left unexamined). The decision maker only observes Bernoulli feedback (governed by the underlying attraction probabilities) for the items that were examined (including the one that was ultimately clicked upon). Note that a user may potentially scan through an entire list of $K$ items and click on none; this would be denoted by a $K$-dimensional vector of all $0$'s. The goal of the decision maker is to maximize the total number of clicks (in expectation) over $n$ rounds.

For the aforementioned problem, the authors study the regret minimization problem where the learner competes against an oracle that always plays an action (an ordered list of $K$ distinct items) that maximizes the click probability.

The paper proposes matching problem-independent (worst-case) upper and lower bounds (the former based on a UCB approach) for the tabular version of the problem described above. For the linear setting, the authors propose an upper bound (again, using a UCB-based approach) that improves upon the current best known.

**Questions:**

For the tabular case, the bounds match in $n$ and $L$ (up to log factors). Would it be possible to capture explicitly dependencies also w.r.t. $K$?

**Limitations:**

Appears to be well-addressed.

**Strengths And Weaknesses:**

The paper is well-written with a clear technical exposition. The observation that variance-aware algorithms lead to bounds that are order-wise better in this setting than their counterparts relying on fixed variance proxies is an interesting one in light of known results for the standard MAB setting.

I did not evaluate the technical details very rigorously, but proof sketches and arguments appear correct. I reckon this paper makes sound theoretical contributions to this area and would therefore vote for an accept.

---

> ### Author Response · Authors · 2022-08-02
> **response to review**
>
> Thanks for the great question. While the main text only provided the $\tilde{O}$ expression, the appendix provides an explicit bound in the tabular case – just combine the intermediate regret upper bound stated before Lemma 2 with the bounds for $R_1(n)$ and $R_2(n)$ from Lemmas 2 and 3. As a caveat, the constants and logarithms resulting from this explicit bound are not optimized – we wrote the proofs to be as simple as possible, and to unify the analysis in the tabular and linear cases. That being said, the resulting dependence on $K$ is a multiplicative $\log K$ term in the first-order $\sqrt{n L}$ bound, plus an additive term $7 K \log \log n$ which is $\tilde{O}(\sqrt{n L})$ in the nontrivial case $n > L$ (see discussion following Assumption 2 for why this is the nontrivial case).

---

### Meta-Review · Area_Chair_W93P · 2022-08-23

**Recommendation:** Accept
**Confidence:** Certain

**Metareview:**

All the reviewers were generally happy with this paper. There were some comments about a better experimental section and maybe a better discussion of results and extensions (e.g. gap-dependent bounds, what happens in the K->L regime), but everyone felt that the manuscript as written was solid enough to merit acceptance. I encourage the authors to incorporate the discussions on these points in the final manuscript.






**Award:**

No

---

### Decision · Program_Chairs · 2022-09-14

Accept